# Clinical Perspectives for ^18^F-FDG PET Imaging in Pediatric Oncology: Μetabolic Tumor Volume and Radiomics

**DOI:** 10.3390/metabo12030217

**Published:** 2022-02-28

**Authors:** Vassiliki Lyra, Sofia Chatziioannou, Maria Kallergi

**Affiliations:** 1Nuclear Medicine Department, General University Hospital of Larissa, 411 10 Larissa, Greece; 22nd Department of Radiology, Nuclear Medicine Section, Attikon University Hospital of Athens, 124 62 Chaidari, Greece; sofiac@med.uoa.gr; 3Division of Nuclear Medicine, Biomedical Research Foundation of the Academy of Athens, 115 27 Athens, Greece; 4Department of Biomedical Engineering, University of West Attica, 122 43 Egaleo, Greece; kallergi@uniwa.gr

**Keywords:** ^18^F-FDG PET/CT, metabolic tumor volume, radiomics, pediatric cancer, lymphomas, sarcomas

## Abstract

Pediatric cancer, although rare, requires the most optimized treatment approach to obtain high survival rates and minimize serious long-term side effects in early adulthood. ^18^F-FDG PET/CT is most helpful and widely used in staging, recurrence detection, and response assessment in pediatric oncology. The well-known ^18^F-FDG PET metabolic indices of metabolic tumor volume (MTV) and tumor lesion glycolysis (TLG) have already revealed an independent significant prognostic value for survival in oncologic patients, although the corresponding cut-off values remain study-dependent and not validated for use in clinical practice. Advanced tumor “radiomic” analysis sheds new light into these indices. Numerous patterns of texture ^18^F-FDG uptake features can be extracted from segmented PET tumor images due to new powerful computational systems supporting complex “deep learning” algorithms. This high number of “quantitative” tumor imaging data, although not decrypted in their majority and once standardized for the different imaging systems and segmentation methods, could be used for the development of new “clinical” models for specific cancer types and, more interestingly, for specific age groups. In addition, data from novel techniques of tumor genome analysis could reveal new genes as biomarkers for prognosis and/or targeted therapies in childhood malignancies. Therefore, this ever-growing information of “radiogenomics”, in which the underlying tumor “genetic profile” could be expressed in the tumor-imaging signature of “radiomics”, possibly represents the next model for precision medicine in pediatric cancer management. This paper reviews ^18^F-FDG PET image segmentation methods as applied to pediatric sarcomas and lymphomas and summarizes reported findings on the values of metabolic and radiomic features in the assessment of these pediatric tumors.

## 1. Introduction

According to epidemiological data in the U.S and Europe, cancer incidence in childhood is at least 30-fold lower compared to that in adulthood, corresponding roughly to one new case per year over 6500 newborns, children, or adolescents [1,2,3]. Despite its rarity, this implies that one child in approximately 300 will be diagnosed with cancer before their twentieth birthday. Moreover, the childhood cancer incidence rate has been slightly increasing during the past few decades without any known causes [1,2]. As life style risk factors, commonly implicated in adulthood carcinogenesis, can be virtually excluded, older maternal age, higher birth weight, and parental early childhood exposure to environmental factors have been hypothesized as possible causes of increased oncogenic mutations in childhood [4,5]. Not only incidence and causes are different but also cancer types are different between children and adults. Leukemias and brain tumors are prevalent in childhood (~50% of all cases), followed by lymphomas and sarcomas (~25% of all cases) [1,2,3]. Blastomas (embryonal tumors) and germ-cell tumors are common (~12% of all cases), while carcinomas and precancerous dysplasias, which are almost 85% of cancer cases in adults, are exceptionally in childhood [3,6,7,8,9]. Instead of precancerous dysplasias and in situ carcinomas, benign tumors with possibly malignant degeneration are occasionally noticed in rare genetic syndromes, as neurofibromatosis [10]. Sarcomas and blood cancers occur at any age but their histologic subtype is different in children and adolescents than in adults. Moreover, pediatric tumors generally present high histologic grade/aggressiveness, and consequently, intensive treatment regimens are commonly adopted to achieve treatment goals [8,11].

Since the 1970s, pediatric cancer management has observed significant increase in five-year survival (from 58% to 84%) and more than 50% lower overall mortality [12]. As childhood cancer incidence trended upward during this period, significant treatment advances and multimodal approaches led to this important achievement. Advanced imaging modalities for accurate staging, including PET/CT imaging, well-organized assessment (pediatric cancer rarity imposes constitution of cooperative groups for the conduct of clinical trials), and rapid adoption of new treatment strategies by the pediatric oncologists (even for non-enrolled population in clinical trials) have also positively influenced the registered survival rates [12]. However, the price to be paid by the childhood cancer survivors is the increased risk for long-term side effects, more severe than in adults, due to the greater sensitivity of the developing organs and the higher life expectancy [13,14,15]. According to alarming previously published data by the Childhood Cancer Survivor Study (CCSS) in early 2000s, 18.1% of five-year childhood cancer survivors died within the subsequent three decades, revealing a different pattern of late-mortality with time. Specifically, the mortality in young adults from second cancer or cardiac or pulmonary disease related to the previous intensive treatment exceeded that of primary cancer recurrence 15 years post diagnosis [16]. A downward trend in late mortality among childhood cancer survivors, registered by the recently published data of the CCSS, is the first evidence of the effectiveness of the adopted lowering radiotherapy and anthracycline-based chemotherapy in the past two decades [17,18]. Hence, the better the risk stratification of pediatric oncologic patients, i.e., the more selective and accurate, the fewer the under- or overtreated cases.

As risk assessment is the gatekeeper for treatment optimization and personalization, molecular imaging and molecular biology advances are pursuing this goal together. PET imaging, although with a well-validated role only in Hodgkin’s Lymphoma (HL), is diffusely used in a variety of childhood cancers. Childhood cancers have an overall high ^18^F-FDG uptake, mainly correlated with their aggressiveness and their high grade of dedifferentiation. ^18^F-FDG PET/CT or ^18^F-FDG PET/MR hybrid imaging may still need standardization but is very helpful firstly in staging and recurrence detection and secondly in treatment response assessment of a variety of types of lymphomas, sarcomas, blastomas, and germ-cell childhood tumors [19,20,21]. Tumor standardized uptake values (SUVs), e.g., maximum SUV (SUVmax), mean SUV (SUVmean), SUVmax standardized to lean body mass (SULmax), and peak SUV (SUVpeak) in a small, fixed-size, usually of 1 cm^3^, the volume of interest, represent the semi-quantitative parameters of tumor ^18^F-FDG uptake [22,23,24,25]. A prognostic value of the universally used SUVmax has been demonstrated by several studies in a variety of tumors [25,26,27,28,29], but in clinical practice, SUVmax is essential for treatment response assessment. The quantitative PET (qPET) evaluation, defined by the ratio of Lesion-SUVmax with Liver-SUVmean has been recently adopted in pediatric lymphomas due to the demand for better standardization of treatment response assessment [30,31]. This and the straightforward ratio brought higher confidence in distinguishing good mid-treatment responders in pediatric HL [32].

In addition, advances in digital image segmentation and processing led to the availability of large “quantitative” datasets based on spatial imaging decomposition analysis. The recognition of different image-element spatial distribution patterns in tumors, otherwise different tumor “textural” features, could offer the possibility of a more accurate image-based risk stratification and treatment group classification of oncologic patients. This review is an attempt to understand the perspectives of clinical impact for already familiar “manually”-defined tumor features and the recently introduced “artificial intelligence”-defined tumor features on ^18^F-FDG PET imaging in pediatric oncology and on pediatric lymphomas and sarcomas, in particular, where ^18^F-FDG PET imaging is widely used, as shown in the pie chart of Figure 1.

## 2. Metabolic Tumor Volume and Radiomics

As revealed by its name, metabolic tumor volume (MTV) or total metabolic tumor volume (TMTV) refers to the sum of the volume of metabolically active or otherwise ^18^F-FDG avid tumor lesions. Tumor lesion glycolysis (TLG) is the other relational PET parameter defined as the product of MTV and SUVmean [33]. These quantitative parameters offer the possibility of disease burden evaluation in PET imaging. Consequently, their prognostic role has been hypothesized and repeatedly confirmed, although retrospectively, by many PET studies in many types of tumors, in the last two decades [34,35,36,37,38]. Despite promising results, MTV/TLG measurements have not yet been implemented in clinical practice or investigated in systematic trials to confirm their potential role in risk stratification and treatment group delineation of cancer patients. This is mainly related to difficulties and uncertainty regarding their standardization [39,40].

Many different segmentation methods have been proposed to improve reproducibility, standardize MTV calculation, and determine the best “fitted” MTV regions of interest (ROIs) [41]: (a)“Manual” ROI outlines of tumor lesions could be the most accurate method for MTV calculation but they are time-consuming and have low reproducibility due to inter-observer variability.(b)“Fixed threshold”-based methods, as those using SUVmax or SULmax absolute values, for example 2.0, 2.5 or above, are simple and widely used for tumor volume delineation. However, “fixed threshold” selection is arbitrary and presents an inherent risk of excluding low or overestimating high ^18^F-FDG uptake tumor lesions.(c)“Fixed relative threshold”-based methods, as those based on 41% or 50% of tumor SUVmax or SULmax, may also overestimate the volume of tumor lesions close to high background areas, as it is the heart in the mediastinum or otherwise, the volume of low-uptake tumor lesions. Consequently, MTV/TLG calculation could be inaccurate because of intra- and/or inter-lesional SUV variability during initial staging or treatment response evaluation [42].(d)“Adaptive threshold”-based methods represent an effort to adjust the threshold to tumor image characteristics, as according to published data, the threshold should probably be higher when the volume of a tumor lesion decreases after treatment [41,43,44,45].(e)“Gradient” methods are based on the SUV gradient evaluation to offer a tumor FDG uptake-independent method and optimize the less reproducible threshold-based segmentation methods [46].(f)“Fully automated” methods are relatively easy to apply to PET imaging but, unlike the manual or semi-automated processes, their accuracy is seriously challenged by tumor heterogeneity and imaging conditions [47].

It is clear from the reported studies that there is no universally “good” segmentation method. It is highly likely that the optimum technique is tumor specific or time specific, i.e., before or after treatment. For example, HL and Ewing’s sarcoma show quite different “heterogeneity” in lesion uptake or in lesion size both at initial staging and after treatment. According to Im et al. [41], “fixed absolute threshold” segmentation methods could generally be more accurate as survival prognosticators for initial MTV evaluation regardless of the tumor type. In contrast, “adaptive”, “gradient”, or “fully automated” algorithms could generally be more accurate than “fixed relative threshold” methods avoiding treatment response overestimation of highly heterogeneous lesions. 

The concurrent evolution of more advanced segmentation methods and statistical models for image analysis based on “artificial intelligence”, “machine-learning”, and ultimately, “deep-learning” algorithms offer the possibility of simultaneous quantification and clustering of a large amount of tumor imaging data, including MTV and TLG parameters. A key aspect to the clinical implementation of this new information is its reproducibility. 

The use of the new term of “*radi-omics*”, similarly to the terms “gen-omics”, “prote-omics”, or “metabol-omics”, with which it shares the common suffix “-omics”, emphasizes the large number of quantitative image features. A “radiomic” tumor image analysis includes the evaluation of “traditional”, “hand-crafted”, or “first-order” features, concerning the distribution of intensity within the segmentation and “advanced”, “deep”, or “high-order” features [48,49,50,51,52,53,54,55,56,57,58,59]. Although the “nature” of each of these tumor image features is not known, they could be clustered in those of *size* (e.g., area, surface area, maximum 3D diameter, least axis, major axis, MTV/TLG), *shape* (e.g., elongation, sphericity, flatness), and *texture*, including *textural feature families* (e.g., intensity histogram, gray level co-occurrence matrix, neighboring gray level dependence matrix, gray level run length matrix, gray level size zone matrix) and their individual *textural features* (e.g., 10th/90th percentile, skewness, kurtosis, energy, entropy, contrast, coarseness, homogeneity, dissimilarity, non-uniformity, emphasis) [48,49,50,51,52,53,54,55,56,57,58,59]. The *textural* image features, based on the statistical model elaboration of spatial and pattern distribution of pixel tumor data, most probably include a lot of useful information. The power of prognostication of tumor textural features or, as more appropriately suggested, of the tumor “radiomic signature” [48], could be more than a hypothesis, as revealed by the first evidence data [49,50,51,52]. In addition, recently introduced “*radio-genomics*” approaches highlight the effort to correspond radiologic with genomic characteristics to decode tumor phenotype noninvasively. In particular, “radio-genomics” could meet the need for further image-based risk stratification during the initial assessment of cancer patients, offering the advantage of the entire tumor mass evaluation and not only of the biopsy specimen [60,61]. Regarding the latter, there is a large volume of published data regarding the role of ^18^F-FDG PET/CT imaging in directed biopsy of many types of tumors, as the distribution of ^18^F-FDG uptake in tumor mass could guide the detection of the most aggressive component [62].

Tumor “*heterogeneity*” is considered a significant predictor of treatment failure but it is difficult to quantify objectively and consistently. It is usually qualitatively described through visual assessment of the images [54,55]. Heterogeneity increases with tumor volume and decreases with response to treatment. Consequently, it could be used for risk stratification as well as treatment response assessment. The high sensitivity of ^18^F-FDG PET/CT imaging in detecting tumor heterogeneity could be better expressed by the several quantitative parameters of tumor textural analysis instead of the SUV variation analysis [56,57]. Moreover, tumor heterogeneity, as expressed by textural analysis, could differ due to different histopathologic subtypes or different clones or other tumor type and host-related differences, which could have a significant clinical impact in pediatrics.

The use of a specific image feature in clinical practice, even without knowing its genomic or molecular meaning, depends on the reproducibility and robustness of this feature through serial images of the same patient or through different patients and scanners [52,53,54,55,56,57,58,59]. The rapidly evolving tumor digital decoding field into quantitative features of size, shape, and texture is highly complicated, and consequently, much more influenced by all these parameters that could influence SUV quantification in PET tumor images. The more standardized the image acquisition protocols are, the more reliable the PET “radiomic analysis” will be. As acquisition protocol uniformity is not feasible, the parameters to be considered should be investigated to guarantee the most reliable approach. Moreover, as “radiomic” analysis is segmentation-depended, the standardization of the segmentation process is most critical for the reproducibility of the extracted data. According to what has been mentioned previously about MTV evaluation, the semi-automated or automated segmentation processes are generally more objective but semi-automated approaches yield better results in low or heterogeneous tumor uptake or lower image quality.

Another problem that has to be resolved is the amount of input data. Deep learning algorithms require the evaluation of large amounts of input data in order to be well “trained” for accurate and reproducible tumor analysis that could be used in risk stratification of oncologic patients [56,57,58,59]. As pediatric cancer is rare, image-based precision medicine in pediatric oncology depends primarily on the extensive cooperation of international groups.

## 3. Pediatric Sarcomas

Pediatric bone and soft-tissue sarcomas account for almost 12% of malignancies in children and adolescents, with an incidence of over 300 per million per year under the age of twenty [2,63]. Contrary to adult sarcomas, the prevalent pediatric types are osteosarcomas (OST) and Ewing sarcomas (ESFT, Ewing sarcoma family tumors) in adolescents and rhabdomyosarcomas (RMS) in preschoolers [9,63]. 

At present, ^18^F-FDG PET/CT imaging is widely used in the staging of pediatric sarcomas. However, in the protocol of the current international randomized trial ΕuroEwing 2012, amended on June 2017 [64,65], ^18^F-FDG PET/CT is still not mandatory and has not replaced the bone scan during the initial assessment of ESFT. Several studies have highlighted the high sensitivity of ^18^F-FDG PET/CT in revealing bone/bone marrow metastases in sarcomas, particularly in ESRT and RMS [66,67,68,69,70,71,72,73]. These studies, although retrospective and with a small number of pediatric patients, suggest that ^18^F-FDG PET scan can reveal more skeletal metastatic foci than the bone scan except for skull metastases. Another objection exists regarding osteosclerotic lesions, not FDG avid and without soft-tissue component, but this is rarely the case during ESFT initial staging [68]. Hence, based on the high sensitivity and, consequently, the high negative predictive value of the ^18^F-FDG PET/CT scan, the bone scan could be safely omitted during initial assessment of the Ewing sarcoma [70]. The accumulated experience from the empirical use of PET scans and the abovementioned study results [66,67,68,69,70,71,72,73] have been considered in the protocol recommendations of the new RMS randomized trial [74], paving the way for a more widespread use of PET imaging in pediatric sarcomas. In order to achieve better risk stratification, the PET scan has become mandatory, while the bone scan should only be performed if the PET scan is not available. The role of ^18^F-FDG PET in baseline risk stratification of RMS tumors has also been highlighted by the new protocol recommendations due to its ability to identify suspicious metabolically active lymph nodes for further investigation [74,75]. 

Based on the above, ^18^F-FDG PET scan should be used during the initial assessment of sarcomas to investigate bone and lymphonodal metastatic disease, in particular [20]. Metastatic disease is the most important prognostic factor of poor disease outcome and, in the case of bone metastases, the number of sites is still essential for risk substratification of pediatric patients with primary disseminated multifocal Ewing sarcoma (PDMES) [76]. However, even in localized tumors, pediatric sarcomas, such as ESFTs, are considered at high risk for dispersion or coexistence of micrometastatic disease and should be addressed as systemic [65]. The evaluation of minimal disseminated disease, already integrated as additional information in the revised staging system of pediatric non-Hodgkin lymphomas (NHL) [77], is currently under investigation even in sarcomas [78,79,80]. Tumor bulk (>200 mL) is the second most important poor prognostic factor in pediatric sarcomas [81], as in pediatric lymphomas [77]. The EuroEwing Consortium (EEC) has proposed the following formula for more standardized tumor volume (TV) evaluation in the trial-enrolled population [64]: TV = a × b × c × F, where a, b, and c represent the maximum tumor dimensions in three planes, with F = 0.52 for spherical tumors and F = 0.785 for cylindrical tumors. The third most crucial survival prognostic parameter is the response to induction chemotherapy, and, in the case of resectable tumors, it is based on the assessment of histopathological response. Good responders are considered those with viable tumor cells <10% in the post-surgical specimen [64,65]. This means that tumor volume reduction evaluated by response evaluation criteria in solid tumors (RECIST), or similar conventional imaging criteria, cannot predict histopathological response [82,83,84]. Radiological response (tumor volume regression with a cut-off value of 50%) is only used for unresectable tumors, in which pathology could not be predictive of survival and local control is based on radiotherapy [65]. 

Regardless of the abovementioned risk stratification parameters, the outcome of pediatric sarcomas is not comparable to those of lymphomas. It has unfortunately remained stationary at low levels during the last two decades, with a 5-year event-free survival (EFS) <65% and <35% for good and poor risk ESFTs, respectively [63,64]. It seems that, contrary to pediatric lymphomas (5-year EFS > 90%) [77], where the efforts should focus mainly on better risk stratification for lowering overtreated cases, the need for new highly efficient and not excessively toxic agents is probably more imperative in sarcomas [11]. Risk stratification, important in predicting survival, is less powerful in the management of oncologic patients when treatment options are limited. In the imminent case of treatment advances, like those in the field of targeted therapy [85], resolution of risk stratification issues in pediatric sarcomas should lead to proper and more effective use of new agents. Interestingly, pretreatment measurements of SUVmax and SULmax of bone and soft-tissue sarcomas suggest that these metrics may be independent survival prognostic factors [28,29,86,87,88,89,90]. 

A literature search has been performed in PubMed, Embase, and the Cochrane Library for a 5-year period from 2016 to 2021. The primary search was restricted to title and abstract and subsequently, to full-text of the studies (eight studies). Two older studies (one of them listed in Table 1) were selected by checking cross references. All the important findings of this search are listed in Table 1 and summarized in the following paragraphs. 

A recently published study [91] in a small sample of 34 pediatric patients with OST, most of which with localized osteoblastic osteosarcoma, has prospectively confirmed the prognostic value of SUV and other PET metabolic parameters (Table 1). When dichotomizing SUVpeak, a statistically significant correlation was revealed with EFS and overall survival (OS) for all the evaluation time-points (baseline, mid-treatment, and end-treatment of neoadjuvant or induction chemotherapy). SUVmax was correlated with survival but less significantly than SUVpeak, probably reflecting the higher variability of the single pixel-based SUVmax value [23]. Similarly, the metabolic volume parameters MTV/TLG calculated by fixed “absolute” (SUV, 2.0 and 2.5) or “liver-based” (liver SUVmean+2SD, standard deviations) thresholds (in general, in the range of SUV, 3–4) [41], demonstrated a statistically significant correlation with survival at all time-points of the primary follow-up. “Liver-based” MTV/TLG values were independently correlated with EFS, even after adjusting for the survival predictors of stage and histopathological response. On the contrary, the MTV/TLG-fixed “relative” (40% and 60%) based threshold values were inaccurate in predicting outcome, probably due to MTV overestimation related to the method’s limitations in case of the lower and more heterogeneous post-treatment uptake [91]. A percentage change of MTV/TLG metabolic parameters (ΔMTV/ΔTLG), as a percentage change of SUV (ΔSUV) between baseline and post-treatment values, were also statistically significant correlated with survival, although they did not reveal a stronger correlation compared to that of baseline parameters [91]. However, in a most recent, although retrospective, study, in patients with localized ESFT [92], ΔTLG revealed the best prediction for histopathological response with 100% sensitivity and 77.8% specificity. Finally, it is important to mention that when dichotomizing end-treatment (post-induction chemotherapy) MTV/TLG values, calculated by fixed “absolute” and “liver-based” thresholds [91], a correlation with good or poor histopathological response was revealed, in agreement with another prospective study, most recently published on pediatric ESFT patients [93] (Table 1). 

More importantly, in two older prospective studies [94,95], based on a similar population of pediatric osteosarcomas, the residual metabolic volume and activity of sarcomatous lesions after a single course of induction chemotherapy, evaluated by “fixed absolute thresholds”, were early predictors of tumor necrosis fraction, i.e., of histopathological response to treatment. 

In addition, textural analysis has also been most recently investigated in a prevalently pediatric sarcoma population [96,97,98] (Table 1), as a way to obtain an imaging “point of view” of tumor heterogeneity implicated in tumor chemosensitivity and response to treatment [54]. In the study of Bailly et al. [96], no correlation was found between radiomic, or even metabolic data, extracted from baseline tumor PET images of children with ESFT and OST using “adaptive” segmentation and histopathological treatment response and outcome; the only exception was the tumor “textural” shape feature of “elongation”. In contrast, in the study of Jeong et al. [97], a percentage change between baseline and post-induction chemotherapy of few PET “textural” features, selected based on their higher reproducibility [62], could accurately predict histopathological response. After combining the principal component analysis method with a “trained” machine learning approach using data from the OST pediatric patients, even a baseline PET tumor textural feature became more accurate in predicting good and poor responders to induction chemotherapy [97]. Similarly, a more advanced “trained” deep learning approach, based on two-dimensional convolutional neural networks (2D-CNN), could further improve prediction of treatment response from several baseline textural features [98]. These results reflect the interference of training data and the segmentation method and processing, with the peculiarities of the different textural features.

The reported findings in pediatric sarcomas follow mostly the first evidence data concerning the prediction of response to neoadjuvant chemotherapy by PET metabolic and radiomic features in carcinomas, the typical histologic type of adulthood solid cancers [99,100,101]. In comparison, sarcomas are rare in adulthood; they develop prevalently in soft-tissue, and include more different histologic subtypes than pediatric sarcomas. Thus, the published ^18^F-FDG PET data about MTV/TLG and textural features in adult population sarcomas mainly focuses on identifying the grade of differentiation and distinction of high-grade sarcomas from more common low-grade sarcomas [29,102] and benign lesions [103]. The reviewed retrospective studies in mixed study population samples [97,98] demonstrated that baseline MTV and tumor heterogeneity parameters, such as “nonuniformity” or “coarseness”, could be predictors of histopathological response to neoadjuvant chemotherapy in OST and ESFT patients. In the study by Song et al. [104], baseline MTV was a stronger predictor of survival compared to baseline tumor heterogeneity (Table 1).

**Table 1 metabolites-12-00217-t001:** Summary of observational ^18^F-FDG PET/CT studies, mentioned in the text, including pediatric patients and concerning the predictive value of ^18^F-FDG metabolic and radiomic parameters in prevalently localized sarcomas.

1st Author, [ref]	Year	Study Design	Cancer Type	Population (Mean/Median Age)	^18^F-FDG PET/CT Time-Points	^18^F-FDG Parameters Correlated with Prognosis	Segmentation Methods (Thresholds)	Prognostic Parameters Predicted
Li Y-J., [90]	2016	Meta-analysis P:8/R:15	B & STS	1261 *	Baseline, post-NAC	SUVmax, MTV, TLG	NR	EFS, OS
Im HJ., [91]	2018	P	OST	34 (12.2)	Baseline, interim, post-NAC	SUVpeak, MTV, TLG	Fixed-absolute and liver-based	EFS, OS Histologic response
Annovazzi A., [92]	2021	R	ESFT	28 (28.7) *	Baseline, post-NAC	ΔTLG (cut-off: −60%)	Fixed-relative (40% SUVmax)	Histologic response
El-Hennawy G., [93]	2020	P	ESFT	36 (9.6)	Baseline, post-NAC	MTV2_(L)_ (cut-off: 17 mL) TLG2_(L)_ (cut-off: 60 g) ΔTLG_(L)_ (cut-off: −90%)	Fixed-absolute and liver-based	Histologic response
Byun BH., [94]	2014	P	OST	30 ** (NS) [17 ≤ 15 years 13 > 15 years]	Baseline, interim, post-NAC	MTV2.5 (interim) ≥ 47 mL TLG2.5 (interim) ≥ 190 g	Fixed-absolute (SUVmax: 2.0 and 2.5)	Histologic response
Bailly C., [96]	2017	R	OST, ESFT	61 (13.9)	Baseline, post-NAC	Elongation (shape textural feature) †	Adaptive	EFS, OS for OST
Song H., [104]	2019	R	OST	35 (33) *	Baseline post-NAC	MTV and radiomics (LA, DNU, GLRL_NU, GLSZ_NU)	Manual (ITK-SNAP 3.8.0)	EFS Histologic response
Jeong SY., [97]	2019	R	OST	70 * (NS)	Baseline, post-NAC	MTV, TLG, and radiomics (LCM_Entropy)	MLA	Histologic response
Kim J., [98]	2021	R	OST	105 ** (NS) [80 ≤ 19 years 25 > 19 years]	Baseline post-NAC	MTV, TLG, and radiomics (GLCM_entropy, GLSZM_HGLZE GLRLM_SGHGE, NGLDM_SNE)	MLA DLA (2D-CNN)	Histologic response

* Mixed population, prevalently adult; ** Mixed population, prevalently pediatric; † SUVmax, SUVpeak, MTV, and TLG did not correlate with survival or histologic response to NAC; P (prospective study), R (retrospective study), B and STS (bone and soft-tissue sarcomas), OST (osteosarcomas), ESFT (Ewing sarcoma family tumors), NS (not specified), post-NAC (post-neoadjuvant chemotherapy), SUVmax (maximum standardized uptake value), SUVpeak (peak standardized uptake value), MTV (metabolic tumor volume), TLG (tumor lesion glycolysis), ΔTLG [differential TLG: (baseline TLG—post-NAC TLG/baseline TLG) × 100%], MTV2_(L)_ (post-NAC MTV estimated by liver-based threshold), MTV2.5 (MTV estimated by fixed absolute threshold of SUVmax = 2.5), LA (least axis), DNU (dependence non uniformity), GLRL_NU (Gray Level Run Length_NonUniformity), GLSZ_NU (Gray Level Size Zone _NonUniformity), GLCM_Entropy (Gray Level Co-occurrence Matrix _Entropy), GLSZM_ HGLZE (Gray Level Size Zone Matrix_High Gray Level Zone Emphasis), GLRLM_SGHGE (Gray Level Run Length Matrix_High Gray Level Run Emphasis), NGLDM_SNE (Neighboring Gray Level Dependence Matrix_Small Number Emphasis), NR (not reported), MLA (machine learning approach), DLA (deep learning approach), EFS (event free survival), OS (overall survival).

Although multistep reproducibility issues should first be addressed [52,53,54,55,56,57,58], the limited first evidence data in the pediatric sarcomas listed in Table 1 support a more extensive clinical implementation of the ^18^F-FDG PET tumor metabolic and radiomic parameters and allow the following observations:(a)The method of MTV evaluation should “join” the clinical context, otherwise, the type of sarcoma and time of evaluation. OST patients, in general, have lesions with less soft-tissue component and consequently less post-treatment volume shrinkage than ESFT patients. Moreover, persistent bone ^18^F-FDG uptake could be related to the post-treatment bone-healing reaction [92]. Thus, MTV-fixed “relative” methods could preferably be avoided to limit post-treatment MTV overestimation in OST patients [92].(b)An early prediction of histopathological response by MTV/TLG and textural features could be most useful after approval of new targeted therapies, which aim to change the standard of care and outcome for pediatric sarcomas. In the current published guide for the practical evaluation of PET response criteria in solid tumors (PERCIST) [105], the concomitant estimation of MTV/TLG parameters (usually by liver-based threshold segmentation methods) has been proposed for better consensus in the assignment of stable, partial, or progressive response to induction treatment. However, reproducibility of MTV/TLG evaluation is a prerequisite for treatment response prediction, still interfering with the MTV/TLG prognostic value in clinical practice.(c)It should be clear that given the histologic type, pediatric sarcomas are different compared to those of adults. The tumor microenvironment is much more important and a possible target for immunotherapy agents, as implicated in the tumor response to treatment. On the contrary, mutational load and relative neoantigens are less expressed by tumor cells of pediatric sarcomas compared to adult sarcomas. Thus, targeted agents, as those implicated in cell differentiation, are probably more effective in pediatric sarcomas, according to experimental data for the “embryonal” RMS histologic subtype, the most common soft-tissue pediatric sarcoma [85,106,107,108]. Overall, tumor heterogeneity ^18^F-FDG imaging data reflects the histologic subtype, tumor microenvironment, and tumor molecular and genomic characteristics. Integrating all this information could lead to a more accurate interpretation of PET-based risk stratification and treatment monitoring of the whole tumor lesion of pediatric sarcomas. Interestingly, the first data in “radiogenomics” of adulthood carcinomas revealed an accurate tumor phenotyping and decoding of breast cancer lesions by PET/MR textural features [109,110].

Concerning RMS, adolescents, regardless of the histologic subtype (embryonal or alveolar), have a worse prognosis than preschoolers [111]. As previously mentioned, dichotomizing the patient population into children and adults is useful but arbitrary. In the context of textural features’ cut-off values and interpretation, such as the shape feature of elongation in pediatric OST [96], age and other somatometric parameters could contribute to a better understanding of the biological characteristics of pediatric sarcomas.

In conclusion, treatment response evaluation by radiomic analysis should include both primary and metastatic tumor sites if the minimum required volume is guaranteed [54]. Volume-dependent [57] radiomic analysis that estimates the dynamic process of tumor heterogeneity could help guide decisions for further local treatment of metastatic sites. Inversely, radiomic analysis could detect recurrence in pediatric sarcomas or even sarcomatous dedifferentiation of neurofibromas in children with neurofibromatosis type I more accurately than the SUV indexes. Finally, PET-directed biopsy and in the case of unresectable sarcomas, PET-based radiation treatment planning [112], could represent the further application fields of radiomic analysis in heterogeneous pediatric sarcomas. Evidently, multiparameter radiomic analysis, including CT or MR imaging systems, could be integrated into ^18^F-FDG PET metabolic and textural parameters and interchanged between the two different PET/CT or PET/MR hybrid systems used in pediatric oncology [113] for even better risk-stratification, treatment response prediction [114], and recurrence detection of pediatric sarcomas.

## 4. Pediatric Lymphomas

Pediatric lymphomas constitute about 12% of pediatric malignancies. HL are less frequent compared to NHL and account for about 40% of pediatric lymphomas [3,7]. HL probably represents the most common indication for ^18^F-FDG PET/CT imaging in pediatric oncologic patients (Figure 1), reflecting that the role of PET imaging in HL is mostly standardized and, more significantly, is used to implement a modified therapeutic strategy. Unlike pediatric sarcomas, the treatment response assessment in pediatric lymphomas is PET-based. As mentioned above, as the survival rate achieved by treatment advances in lymphomas, especially in HL, is exceptionally high (>95%), the new challenge is to reduce overtreatment [115]. Thus, unlike HL treatment management in adults, radiotherapy, particularly toxic in underage patients, should be administered even more selectively. Indeed, the previous multicentric trial for the classical HL, Euro-PHL-C1 [32], leads to two crucial achievements. The first one was the optimal outcome, despite omitting radiotherapy in good mid-treatment responders or otherwise, in children with the interim PET of Deauville score (DS) 1 and 2 [116]. The second one was the validation of qPET, although retrospectively in the Euro-PHL-C1 trial population, which was a more reproducible semiquantitative method for treatment response evaluation than the Deauville score [31]. Based on these two previous achievements, the ongoing multicentric trial, Euronet-PHL-C2 [117], is currently focused on the achievement of further radiotherapy reduction, avoiding interferences with curability. For that reason, early-stage HL pediatric patients with risk factors, including tumor bulk ≥ 200 mL (calculated as ellipsoidal volume, by the product of the principal axes divided by 2 [117]), have been upgraded from chemotherapeutic treatment level 1 (TL1) to chemotherapeutic TL2. In contrast, the indications for post-chemotherapeutic irradiation have been reduced. In particular, according to the new protocol [117], radiotherapy is omitted in a wider pediatric patient population, as good mid-treatment responders are considered children with a qPET score < 1.3, which approximately corresponds to children with a Deauville score of 1, 2, and 3. Moreover, some ^18^F-FDG PET avid lesions, as small lymph nodes with the largest diameter < 10 mm, are not included in initial or residual disease burden, which means that these lesions will not be irradiated in the case of inadequate response. In addition, bone marrow involvement is based only on ^18^F-FDG PET findings, excluding biopsy and molecular assessment techniques to evaluate minimal disseminated disease (MDD). This is also following the overtreatment reduction strategy of the ongoing trial [117], as the omission of minimal disseminated disease, similar to that of small lesions, is considered without effect on TL upstaging or the prognosis of pediatric HL patients.

Pediatric NHL is generally considered a diffuse disease regardless of typical staging, based on conventional or ^18^F-FDG PET/CT imaging. Thus, MDD assessment in bone marrow or peripheral blood, by FISH (fluorescence in situ hybridization) or PCR (polymerase chain reaction) analysis, is reported as additional information in the revised international pediatric NHL staging system (IPNHLSS) [77]. Although stage-dependent, MDD is considered an additional tool for the prognosis and evaluation of response to treatment [118]. Both PET/CT imaging and molecular techniques (FISH and PCR) for the assessment of minimal residual disease (MRD) are included in the international response criteria for pediatric NHL [119]. As about 40% of lymphomas present residual masses after treatment [119], ^18^F-FDG PET/CT and qPET evaluation is preferred. However, the clinical impact of qPET evaluation or otherwise, of PET-negative or PET-positive residual masses, is still unclear in pediatric NHL, as opposed to HL. According to published data, the negative predictive value of post-treatment PET findings in pediatric NHL is high but the positive predictive value is low [120,121,122]. Therefore, metabolically active residual masses, particularly of qPET around 1.3, should be further investigated by biopsy for differential diagnosis between residual disease and post-treatment inflammation [77]. Moreover, the refinement of the Lugano classification response criteria for adulthood NHL in the era of immunomodulatory agents highlights another issue [123]. The tumor “flare” phenomenon should be considered to avoid false-positive results during PET-based treatment response assessment. These agents have been already integrated in the management of relapsed/refractory pediatric lymphomas [124] and are being increasingly integrated in the front-line treatment of advanced pediatric lymphomas, accordingly, in the recently published results of a randomized and international trial [125]. As an initial or salvage therapy, chemotherapy is the primary treatment in pediatric NHL, unlike pediatric HL, where radiotherapy could be integrated into the therapeutic protocol template [117]. However, despite the current high-efficiency first-line treatment strategies and the optimal response and outcome achieved in the overwhelming majority of pediatric lymphoma patients, a small amount of about 10% of the cases, concerning mainly NHL patients, will not respond or will relapse, significantly compromising the outcome [126]. Consequently, treatment optimization and precision medicine in pediatric lymphomas have to deal with primary identification of overtreatment or undertreatment of cases to adequately and timely reduce or intensify a first-line treatment approach. PET metabolic and textural parameters could enhance the accuracy of PET imaging in the management of pediatric lymphomas.

Unlike sarcomas, the pretreatment SUVmax is generally not considered a prognostic factor in lymphomas [27,127,128]. SUVmax is generally lower in HL than NHL, probably related to the higher ratio of inflammatory cells in the lymphomatous lesions. Higher SUVmax has been inconsistently related to good and poor prognosis of more proliferative/aggressive lymphomas, due to higher chemosensitivity and a higher propensity to relapse, respectively. It is comprehensible that due to the multifocal/systemic nature of lymphomas, as opposed to sarcomas, SUVmax is less critical than the number of sites and the total volume of lymphomatous disease [129].

A literature search has been performed on pediatric lymphomas, as in the case of pediatric sarcomas. The most important findings of this search are summarized in Table 2 and the following paragraphs.

Despite some contradictory results, especially regarding indolent forms of lymphomas [130,131,132], a recently published meta-analysis, including 24 retrospective and three prospective studies and more than 2700 patients, has revealed the prognostic survival value of baseline MTV/TLG in adulthood HL and NHL [133] (Table 2). Although the cut-offs were method-dependent, MTV/TLG prognostication value was method-independent in both types of aggressive lymphomas. The parameters that tested the skills of segmentation methods were the multiplicity of nodal and/or extranodal tumor sites and the heterogeneity, in the case of the bulky disease, in particular. Despite the known limitations in low uptake or heterogeneous lesions, the negative and positive predictive values of the 41% threshold method were higher than the other methods in the meta-analysis mentioned above [133]. According to the authors, however, as qPET treatment response evaluation in lymphomas is based on liver SUVmean, the liver-threshold methods are theoretically more appropriate for baseline and post-treatment MTV/TLG evaluation in lymphomas. This is because they have the advantage of being adapted to each patient without the percentage-based methods’ variability issues, as the same SUV threshold is applied in all VOIs [133].

Similarly, another most recently published meta-analysis [134], including 41 studies, confirmed the MTV’s prognostication value and the prognostication value of a few radiomic parameters, evaluated in a minority (7/41) of studies, despite the extreme heterogeneity of cut-off values. In addition, in the SAKK38/07 study cohort [135], MTV, evaluated using a fixed SUV threshold of 2.5 and associated with the textural parameter of metabolic heterogeneity (MH) and evaluated by the cumulative SUV volume histogram of the lesion with the highest ^18^F-FDG uptake, could further stratify the outcome in diffuse large B-cell lymphoma (DLBCL). A shorter progression-free survival (PFS) was revealed in DLBCL patients with high MTV and MH values, independent from their response to therapy. Similarly, a correlation between MTV and survival, independent from response to treatment, has also been recently reported in the REMARC study DLBCL cohort population [136]. The role of MTV/TLG in risk stratification could be strengthened by the fact that the current staging systems have a weak correlation with metabolic tumor burden: one-third of those recognized as advanced-stage have the low burden, and, vice versa, about half of the intermediate-risk patients have high tumor burden [137]. Therefore, optimum performance cut-offs are the sine qua non for use in clinical practice, meaning further work is required concerning the methodological and software choices that need to be standardized and simplified. As less variable and mainly less time-consuming, automated segmentation methods are a valuable segmentation option for lymphomas [134,138].

In agreement with previous results [139], a post hoc analysis from the PETAL trial confirmed that baseline MTV with a cut-off value of 328 mL evaluated by the segmentation threshold method of 41% SUVmax could further stratify prognosis in patients with DLBCL in association with interim PET [140]. Similarly, another study based on the early-stage HL population of the H10 trial [141] provided that MTV is critical in the substratification of these patients regardless of the current staging systems. Baseline MTV with a cut-off value of 147 mL evaluated by the 41% SUVmax threshold method improved the predictive value of interim PET treatment response evaluation, classifying patients in those with particularly low, particularly high, intermediate-low, and intermediate-high risk to relapse. High MTV-poor PET interim responders had a 5-year PFS of 25% compared to 95% of low MTV-good PET interim responders [141].

Baseline metabolic and textural parameters were also investigated recently for assessing interim PET treatment response [142]. The investigators studied a cohort of patients with bulky HLs and NHLs with high optimal MTV cut-off value (600 mL). Optimal cut-off values for MTV are influenced by the data acquisition protocol and resolution, the segmentation method, and the study population. The investigators opted to focus on bulky lymphomas, as textural analysis is volume-depended, and consequently, the number of processing data and results increases as lesion volume increases [54,55,56,57]. “Contrast”, “dissimilarity”, “granularity”, and shape parameters (“surface extension” and “2D and 3D fractal dimensions”) were independent predictors of early metabolic response. Although the corresponding value of each of these parameters is only hypothesized (e.g., a higher “surface extension” could be associated with better exposure to the chemotherapeutics) [142], the textural analysis could be another tool for identifying particularly high-risk patients, when the MTV/TLG metabolic parameters are borderline.

Regardless of the reproducibility problems that prevent specific clinical implementation, MTV/TLG have been demonstrated to be prognostic biomarkers in adulthood lymphomas [133,134,135,136,137,138,139,140,141,142,143] (Table 2). However, the following considerations should be considered in the case of pediatric lymphomas. Firstly, the histologic subtypes are usually different. For example, in classical HL, the more common pediatric form in Europe and the United States is nodular sclerosis instead of mixed cellularity, which is more common in adults. Furthermore, the indolent forms of NHL, as a marginal zone lymphoma, are rare in children, in whom the more common subtype is Burritt lymphoma, an aggressive B-cell lymphoma, as opposed to DLBCL, which is the more common subtype in adults. Secondly, children with NHL, as opposed to adults, typically have an extranodal disease involving mediastinal structures, gastrointestinal system, central nervous system, or bone marrow. Thus, apart from the different implications in staging and prognosis in extranodal disease, MTV evaluation may become more challenging than that of nodal disease. Thirdly, PET-based response criteria could differ. Moreover, unlike adulthood NHL, chemotherapy is the only treatment in childhood NHL, as radiotherapy is only exceptionally used. Finally, additional effort is needed in pediatric lymphomas to organize well-conducted studies, due to more important patient number limitations.

A single-center, recently published, retrospective study [144] on 46 pediatric patients with B-NHL investigated if MTV could identify those children most at risk of treatment failure and disease progression. In multivariate analysis, MTV and TLG outperformed serum lactate dehydrogenase and bone marrow involvement on biopsy, confirming that they are predictors of survival. Moreover, baseline MTV and TLG were particularly powerful in substratification of high-risk patients, according to the FAB/LMB 96 risk classification criteria [145], identifying the group of children with particularly high risk for a refractory or relapsed disease [144,146]. Similarly, baseline MTV was the independent prognostic parameter of progression-free and overall survival in pediatric Burkitt [147], LBL (lymphoblastic) [148], and ALCL (anaplastic large cell lymphoma) patients [149]. Results from the AHOD0031 study cohort of pediatric HL patients are also consistent with the fact that the incorporation of baseline MTV into risk-based treatment algorithms may improve outcomes in intermediate-risk (limited stage with risk factors) HL [150]. Importantly, the investigators compared the accuracy of 15 different segmentation thresholds for the MTV/TLG evaluation to predict outcome. After adjustment for the other risk factors, the MTV evaluated by a blood pool-based threshold was the only independent prognostic parameter in pediatric intermediate-risk HL patients [150,151].

Another recently published study [129] on 50 pediatric patients with HL treated with EuroNET-PHL-C1 or -C2 protocol investigated the prognostic value of metabolic and textural PET parameters. MTV was delineated by a fixed relative threshold of 41% SUVmax and a fixed absolute threshold of SUV 2.5 and was manually corrected in about half of the patients due to relatively low activity and high heterogeneity lesions. Baseline MTV and TLG were found predictors of EFS and OS. Although cut-off values were different in the different risk-stratification groups (according to the current pediatric HL classification systems [117]), high MTV best predicted the interim PET inadequate response to induction chemotherapy, in both early and advanced stages (cut-offs of 80 mL vs. 410 mL) as well as in the three (based on stage, tumor bulk, and ESR) treatment levels (cut-offs of 80 mL vs. 160 mL vs. 410 mL). High “asphericity”, a textural parameter reflecting the lesion surface complexity was also correlated with the interim PET inadequate response. In a most recent prospective study in a small population of HL pediatric patients [152], four textural parameters, from the textural families of the grey-level co-occurrence matrix and neighborhood grey tone difference matrix were also predictors of interim PET treatment response (Table 2).

The aim is to further evolve the PET-based management of pediatric lymphomas by predicting interim PET treatment response from baseline MTV and textural PET imaging parameters. The integration of low dose CT or MR images of the PET hybrid systems in textural analysis could provide complementary information that could be particularly useful during the PET-based treatment response evaluation of pediatric lymphomas [153]. This is a primary clinical goal, as the appropriate intensification of front-line treatment will reduce cases of radiotherapy consolidation and cases of inadequate end-treatment response with increased risk to relapse and better balancing achieved between toxicity and curability. The recent multicentric and randomized trial in advanced-stage pediatric NHL [125], concerning the integration of immunotherapy in the front-line treatment, was designed to minimized dismal outcomes and increase the treatment response rate, as the significant predictor of survival. Interestingly, following the concept of “personalized” immunotherapy, MTV could help to adjust administered doses in order to achieve a more effective treatment response [154].

**Table 2 metabolites-12-00217-t002:** Summary of observational ^18^F-FDG PET/CT studies in lymphomas, including eight studies in pediatric lymphomas.

1st Author, [Reference]	Year	Study Design	Type of Lymphomas	Population (Mean/ Median Age)	^18^F-FDG PET/CT Time-Points	^18^F-FDG Parameters Correlated with Prognosis	Segmentation Methods (Thresholds)	Prognostic Parameters Predicted
Guo B., [133]	2019	Meta-analysis P:3/R:24	HL:3 DLBCL:16 Other NHL:8	2729 *	Baseline	MTV, TLG	Fixed-absolute, liver-based, fixed-relative	PFS, OS
Frood R., [134]	2021	Meta-analysis R:41	HL:10 DLBCL:31	>4000 *	Baseline	SUVmax, MTV, TLG MH ** (radiomics)	fixed-absolute, liver-based, fixed-relative	PFS, OS
Ceriani L., [135]	2020	P	DLBCL	141 * (59)	Baseline	MTV, MH ** (radiomics) MTVand MH **	Fixed-absolute (SUVmax: 2.5)	PFS, OS
Vercellino L., [136]	2020	P	DLBCL	298 * (68)	Baseline	MTV (cut-off: 220 mL), MTV and ECOG PS	Fixed-relative (41% of SUVmax)	PFS, OS
Mikhaeel NG., [139]	2016	P	DLBCL	147 * (57)	Baseline Interim	MTV (cut-off: 396 mL), TLG MTV and iPET	Fixed-absolute (SUVmax: 2.5)	PFS, OS
Schmitz C., [140]	2020	P	DLBCL	510 * (62)	Baseline, Interim	MTV (cut-off: 328 mL), ΔSUVmax (cut-off: 66%) MTV and iPET	Fixed-relative (41% of SUVmax)	PFS, OS
Albano D., [143]	2019	R	Burkitt	65 * (53)	Baseline End-treatment	MTV (cut-off: 230 mL) TLG	Fixed-relative (41% of SUVmax)	PFS, OS
Cottereau AS., [141]	2020	P	HL (early stage)	258 * (31)	Baseline Interim	MTV (cut-off: 147 mL) MTV and iPET	Fixed-relative (41% of SUVmax)	PFS, OS
Bouallègue FB., [142]	2017	R	Bulky HL and NHL	57 * (52)	Baseline	MTV (cut-off: 600 mL) Shape/texture parameters (radiomics)	Fixed-Relative (30% of SUVmax)	PFS, OS
Zhou Y., [146]	2020	R	HL and NHL	47 (14.8)	Baseline	TLG	Fixed-absolute (SUVmax: 2.5)	PFS
Kim J., [144]	2019	P	B-NHL	46 (7.5)	Baseline	MTV, TLG	Fixed-Relative (41% of SUVmax)	PFS, OS
Xiao Z., [147]	2021	R	Burkitt	68 (7)	Baseline	MTV (cut-off: 550 mL) TLG (cut-off: 2881 g)	Fixed-relative (41% of SUVmax)	PFS, OS
Yang J., [148]	2021	R	LBL	30 (6.5)	Baseline	MTV (cut-off: 243 mL)	Fixed-relative (41% of SUVmax)	PFS, OS
Mathew B., [149]	2020	R	ALCL	50 (8.5)	Baseline, interim	MTV(cut-off: 180 mL) MTV and iPET	Fixed-relative (40% of SUVmax)	PFS, OS
Milgrom S., [150]	2021	P	Intermediate-risk HL	86 (14.5)	Baseline	MTV	Fixed-absolute (SUV blood pool × 2)	PFS
Rogasch J., [129]	2018	R	HL	50 (14.8)	Baseline	MTV, TLG asphericity (radiomics)	Fixed-relative (41% of SUVmax)	PFS, OS iPET
Rodriguez-Taroco MG., [152]	2021	P	HL	21 (12)	Baseline	GLCM (Entropy, energy) NGTDM (coarseness, busyness)	Fixed-relative (41% of SUVmax)	iPET

* Adult population; ** MH (metabolic heterogeneity) of the target lesion: the lesion with the highest ^18^FDG uptake using the area under curve of cumulative SUV-volume histogram method. P (prospective study), R (retrospective study), HL (Hodgkin lymphoma), DLBCL (diffuse large B-cell lymphoma), NHL (non Hodgkin lymphoma), B-NHL (B-cell non Hodgkin lymphoma), ALCL (anaplastic large cell lymphoma), SUVmax (maximum standardized uptake value), MTV (metabolic tumor volume), TLG (tumor lesion glycolysis), iPET (interim PET), ΔSUVmax [differential SUVmax: (baseline SUVmax − iPET SUVmax/baseline SUVmax) × 100%], ECOG PS (eastern cooperative oncology group performance status), GLCM (grey-level co-occurrence matrix), NGTDM (neighborhood grey-tone difference matrix), PFS (progression free survival), OS (overall survival).

## 5. Other Tumors

Langerhans cell histiocytosis (LCH) is a rare hematologic disorder characterized by the proliferation and accumulation of Langerhans-type clonal cells and accompanying inflammatory infiltrate in various organs and tissues. Although rare (<1% of childhood cancers), the disease is about 10 times more frequent in children compared to adults and more frequently affects bones than soft-tissues. Single-site bone disease is the predominant clinical form of pediatric LCH [155] According to previous and current guidelines for the diagnosis and management of pediatric LCH [155,156], skeletal radiography remains the gold standard for skeletal staging and exclusion of multisite disease. Whole-body MRI is still not integrated in clinical practice but probably represents a promising imaging modality for the future [157,158]. Nowadays, neither bone scan nor ^18^F-FDG PET scan is considered an alternative to skeletal radiography. However, the ^18^F-FDG PET scan, instead of a bone scan, which should not be considered for evaluating multisite bone disease, is considered the most accurate tool for detecting bone and soft-tissue lesions (multisystem LCH) and it is strongly recommended during initial staging [156,157]. Most importantly, by evaluating disease metabolic activity, the ^18^F-FDG PET scan is the most accurate tool for evaluating front-line response to treatment and further decision making in the case of a multisite/multisystem disease [159,160]. The MTV/TLG metabolic parameters may have a role in better risk stratification of multisystem LCH but, to our knowledge, there is no relevant published data probably because of the disease’s rarity both in children and especially in adults, and the poor prognosis of young children in the case of multisystem disease with risk organ involvement.

^18^F-FDG PET/CT also has a complementary role in staging and chemotherapy treatment response assessment of pediatric germ cell tumors, such as yolk sac sacrococcygeal tumors of infants or gonadal tumors of adolescents. The site of disease and the age are significant risk factors in germ cell tumors, with extragonadal thoracic disease and puberty having a worse prognosis [161]. Although pediatric germ cell tumors are rare, testicular germ cell tumors represent more than 10% of adolescent malignancies and are the most common malignancy in young adult men. A recently published retrospective study [162] in 51 young adults with testicular germ cell tumors revealed that MTV and TLG metabolic parameters were significant independent predictors of overall survival, suggesting a similar prognostic role even for testicular germ cell tumors in adolescents. However, despite the significant biological overlap, etiopathogenesis of underage germ cell tumors is generally characterized by a more salient role of abnormal developmental pathways and a relative lack of traditional oncogenes, especially in prime childhood [163]. Although better risk stratification is undoubtedly of clinical impact, the need for more appropriate treatment regimens based on differentiation-induced therapies is more imperative for these non-somatic lineage tumors, to avoid long-term adverse effects in young children and adolescents by the chemotherapeutic cytotoxic drugs used in adults [164].

Finally, neuroblastoma is the most common of pediatric blastomas and accounts for about 6% of childhood cancers, usually affecting infants and children under 15 years old. It is extremely rare in adulthood and is generally considered a neural crest embryonal malignancy with adrenal or extra-adrenal localization. The prognosis is varying, being optimal in low-risk patients and severely compromised in high-risk patients (children ≥ 18 months of age with metastatic disease or children with unfavorable histology), particularly in the case of inadequate response to first-line induction chemotherapy [165]. Regardless of risk stratification, the role of ^18^F-FDG is limited in pediatric neuroblastoma, as adrenomedullary tumor cells are particularly rich of type I amine uptake mechanism. Consequently, staging and treatment response assessment is based on findings of the ^123^I- MIBG [Meta-(radioiodinated)-iodobenzylguanidine] scan, while ^18^F-FDG PET/CT is only limited to the detection of highly dedifferentiated ^18^F-FDG avid/^123^I-MIBG negative neuroblastoma [166]. However, growing evidence based on data from the first prospective study [167] suggests that ^18^F-DOPA PET/CT could be implicated in clinical practice in the imminent future by revealing a higher accuracy in neuroblastoma prognosis, staging, and treatment response assessment compared to ^123^I-MIBG scan. In that case, similarly to ^18^F-FDG, ^18^F-DOPA metabolic tumor volume and tumor lesion metabolic activity parameters could improve risk stratification of pediatric neuroblastoma.

## 6. Conclusions

There is a growing interest in using MTV/TLG and radiomic tumor heterogeneity parameters to quantitatively determine metabolically active disease in pediatric oncology. There is also a significant need to improve the ^18^F-FDG PET/CT evaluation of pediatric tumors by introducing new parameters in addition to SUV, with its known—although well-accepted—limitations [151]. Published data are limited but the studies listed in Table 1 and Table 2 demonstrate that standardized MTV/TLG measurements are more likely to be integrated first in clinical practice instead of the more complex and multi-parametric textural metrics. The clinical implementation of the latter requires coordinated research efforts ideally with large and multicentric prospective studies that will adequately address issues of tumor segmentation and threshold values. Finally, new digital high-resolution scanners are currently being developed, guaranteeing image analysis improvement, while pediatric cancer imaging data libraries are becoming publicly available, offering pediatric oncology experts the required data to develop and test advanced software and new metrics for improved disease and treatment evaluation.

## Figures and Tables

**Figure 1 metabolites-12-00217-f001:**
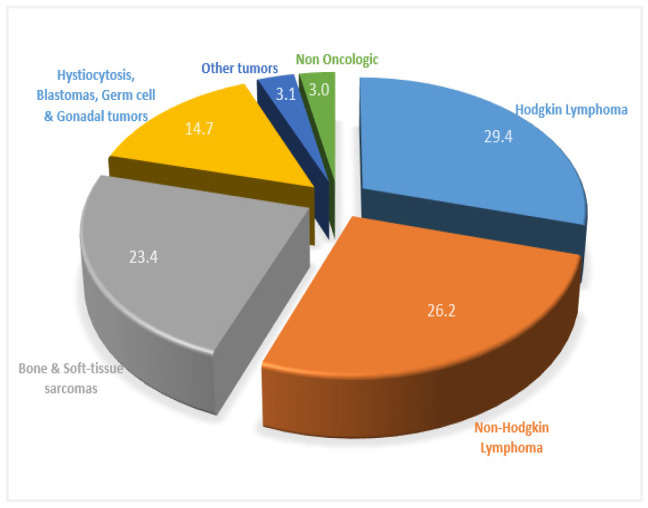
Distribution of ^18^F-FDG PET/CT pediatric examinations based on a 10-year workflow from the Division of Nuclear Medicine, Biomedical Research Foundation of the Academy of Athens (BRFAA). About half of ^18^F-FDG PET/CT exams concern lymphomas, and almost a quarter concerns sarcomas, representing the overwhelming majority of PET imaging in pediatrics.

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
