# Peer review of "Clinical Perspectives for 18F-FDG PET Imaging in Pediatric Oncology: Μetabolic Tumor Volume and Radiomics"

_metabolites, 2022, doi:10.3390/metabo12030217_

Round 1

Reviewer 1 Report

This manuscript is entitled: “Clinical Clinical perspectives for 18F-FDG PET imaging in pediatric oncology: Μetabolic tumor volume and radiomics,” integrated 18F-FDG PET image segmentation methods applied to pediatric sarcomas and lymphomas and summarize the findings of value in assessing metabolic and radiomic features in these pediatric tumors. The authors search that standardized metabolic tumor volume/tumor lesion glycolysis measurements are more likely to be integrated first in clinical practice. Overall, the proposed research is of interest with good potential. The authors carried out detailed studies to prove the concept. There are some questions and suggestions that may need to solve as below:

  1. The authors provide categories of childhood and young people cancers in Figure 1 but do not analyze the tumor’s period. Since cancer metastasis may lead to the development of lesions in different organs, it is essential to know when the symptoms occur or are detected in patients.
  2. The distribution of 18F-FDG PET/CT is available from the analysis of the database provided by the division of nuclear medicine, Biomedical Research Foundation of the Academy of Athens (BRFAA). Can the other medical diagnostic methods also be used to detect these types of cancers? And what are the advantages of 18F-FDG PET/CT over other diagnostic methods? Do different cancer types correspond to more appropriate diagnoses?
  3. Regarding the mention of “SUVmax, SUVpeak, MTV, TLG in the notes to the table did not correlate with survival or histologic response to NAC,” this description should be discussed in the paragraph narrative.
  4. Among the references selected by the authors, there appeared to be inconsistencies among the parameters between the documents, such as population size, age, etc. The differences between the literature and the actual situation need to be mentioned in the discussion.
  5. Whether the data in the table are statistically relevant, such as p-value, standard deviation, etc., such data, if available, should be indicated on the table to provide more statistical confidence.

Author Response

Response to Reviewer 1 Comments

“Overall, the proposed research is of interest with good potential. The authors carried out detailed studies to prove the concept.”

We thank the reviewer for the positive and encouraging comments. Our response to his questions and suggestions is as follows:

Point 1: Extensive editing of English language and style required (but also “the english is correct and readable”).

Response 1: The manuscript was reviewed by a scientist proficient in English and spell check was run on the document.

Point 2: The authors provide categories of childhood and young people cancers in Figure 1 but do not analyze the tumor’s period. Since cancer metastasis may lead to the development of lesions in different organs, it is essential to know when the symptoms occur or are detected in patients.

Response 2: As indicated, Figure 1 represents a “pie chart” of childhood cancers. Each percentage has been calculated by dividing the absolute number of cases registered in Europe in ten years with the total absolute number of childhood cancers within the same period. The data concerning Europe (East, West, North, South) have been extracted from the international registry reports tables. The figure legend has been edited to make it clearer.

Point 3: The distribution of 18F-FDG PET/CT is available from the analysis of the database provided by the division of nuclear medicine, Biomedical Research Foundation of the Academy of Athens (BRFAA). Can the other medical diagnostic methods also be used to detect these types of cancers? And what are the advantages of 18F-FDG PET/CT over other diagnostic methods? Do different cancer types correspond to more appropriate diagnoses?

Response 3: Conventional imaging is obviously used for staging and treatment response assessment of pediatric oncologic patients but as mentioned by the reviewer, the manuscript focuses on the role of PET/CT. However, the role of PET scan compared to conventional imaging is suggested in several points in the manuscript. For example, in the occasion of a) tumor bulk calculation b) inadequacy of conventional imaging RECIST treatment response criteria for solid cancers c) PET-based and not conventional imaging-based decision-making for adjuvant radiotherapy in pediatric Hodgkin lymphoma or inversely d) ambiguous PET-based treatment response assessment in case of residual masses in non-Hodgkin Lymphoma patients.

Point 4: Regarding the mention of “SUVmax, SUVpeak, MTV, TLG in the notes to the table did not correlate with survival or histologic response to NAC,” this description should be discussed in the paragraph narrative.

Response 4: As suggested by the reviewer, we have checked that the parameters correlated with survival, as reported in Table 1, and discussed this in the manuscript.

Point 5: Among the references selected by the authors, there appeared to be inconsistencies among the parameters between the documents, such as population size, age, etc. The differences between the literature and the actual situation need to be mentioned in the discussion.

Response 5: The different characteristics of the studies, such as retrospective or prospective design, population age, types of cancer included, metabolic parameters and cut-off values included, are mentioned in the manuscript in detail for each of the studies reported in the table.

Point 6: Whether the data in the table are statistically relevant, such as p-value, standard deviation, etc., such data, if available, should be indicated on the table to provide more statistical confidence.

Response 6: Our primary intention was to include the statistical parameters mentioned in the related publications but we finally omitted specific values because they complicated the results of the studies listed in the tables due to the variety of the parameters used by the different investigators and the different statistical analyses followed.

Reviewer 2 Report

The paper is interesting and subject is well developed.

Some suggestions/corrections:

  • Ln 26: “genomic” pattern” is not correct.
  • I encourage authors to add a chapter about the predictive value of 18F-FDG metabolic and radiomic parameters in pediatric Hystiocitosis, Blastomas, Germ cell and Gonadal tumors and some other tumors.
  • I suggest completing the title of Table 2 (aligning it with title of Table 1) and adding “(mean age)” to “Population” in Table 2.
  • It is important to follow scrupulously manuscript preparation instructions and to correct any discrepancies (for example in references).

Author Response

Response to Reviewer 2 Comments

“The paper is interesting and subject is well developed.”

We thank this reviewer as well for his positive and encouraging comment. Regarding his suggestions/corrections, we made the following edits to the manuscript:

Point 1: English language and style are fine/minor spell check required.

Response 1: The manuscript was reviewed by a scientist proficient in English and spell check was run on the document.

Point 2: Ln 26: "genomic" pattern" is not correct.

Response 2: As suggested by the reviewer, the phrase “genomic pattern” has been replaced with “genetic profile”.

Point 3: I encourage authors to add a chapter about the predictive value of 18F-FDG metabolic and radiomic parameters in pediatric Hystiocitosis, Blastomas, Germ cell and Gonadal tumors and some other tumors.

Response 3: Thank you for the valuable suggestion. A final section entitled “other tumors” has been added.

Point 4: I suggest completing the title of Table 2 (aligning it with title of Table 1) and adding "(mean age)" to "Population" in Table 2.

Response 4: Thank you for the valuable suggestion. We have added the patients’ mean/median age in Table 2, aligned with Table 1.

Point 5: It is important to follow scrupulously manuscript preparation instructions and to correct any discrepancies (for example in references).

Response 5: We reviewed the references and edited per guidelines.
